# Can the Efficacy of [^18^F]FDG-PET/CT in Clinical Oncology Be Enhanced by Screening Biomolecular Profiles?

**DOI:** 10.3390/ph12010016

**Published:** 2019-01-23

**Authors:** Hazel O’Neill, Vinod Malik, Ciaran Johnston, John V Reynolds, Jacintha O’Sullivan

**Affiliations:** 1Trinity Translational Medicine Institute, Department of Surgery, Trinity College Dublin, D08W9RT Dublin, Ireland; reynoldsjv@stjames.ie (J.V.R.); osullij4@tcd.ie (J.O.S.); 2Department of Radiology, St. James’s Hospital, D08 X4RX Dublin, Ireland; malikvi@tcd.ie (V.M.); cjohnston@stjames.ie (C.J.)

**Keywords:** [^18^F]FDG PET/CT, biomarker profiling, cancer

## Abstract

Positron Emission Tomography (PET) is a functional imaging modality widely used in clinical oncology. Over the years the sensitivity and specificity of PET has improved with the advent of specific radiotracers, increased technical accuracy of PET scanners and incremental experience of Radiologists. However, significant limitations exist—most notably false positives and false negatives. Additionally, the accuracy of PET varies between cancer types and in some cancers, is no longer considered a standard imaging modality. This review considers the relative influence of macroscopic tumour features such as size and morphology on 2-Deoxy-2-[^18^F]fluoroglucose ([^18^F]FDG) uptake by tumours which, though well described in the literature, lacks a comprehensive assessment of biomolecular features which may influence [^18^F]FDG uptake. The review aims to discuss the potential influence of individual molecular markers of glucose transport, glycolysis, hypoxia and angiogenesis in addition to the relationships between these key cellular processes and their influence on [^18^F]FDG uptake. Finally, the potential role for biomolecular profiling of individual tumours to predict positivity on PET imaging is discussed to enhance accuracy and clinical utility.

## 1. Introduction

### 1.1. Positron Emission Tomography

Positron emission tomography (PET) is an imaging modality used in the diagnosis, staging, restaging and monitoring of cancer. It involves the administration of selected labelled molecules that localise in malignant tissues. These molecules integrate into metabolic pathways or act as receptor ligands in cancer cells, concentrating in tumours. Examples include non-specific tracers of metabolism and cell membrane synthesis such as [^11^C]Choline, [^11^C]Acetate and 2-Deoxy-2-[^18^F]fluoroglucose ([^18^F]FDG) and specific tracers such as tyrosine kinase inhibitors that localise exclusively to overexpressed epithelial growth factor receptors on cancer cells [1]. Clinically the most widely used tracer is glucose analogue [^18^F]FDG, a marker of cellular metabolism. Detection of [^18^F]FDG is the basis of functional cancer imaging and is the focus of this review.

### 1.2. 2-Deoxy-2-[^18^F]fluoroglucose ([^18^F]FDG)

[^18^F]FDG undergoes cellular uptake via the same mechanisms as glucose and other hexoses: passive diffusion, sodium dependent transport mechanisms and via specific glucose uptake transporters (GLUTs). GLUTs are expressed on the membranes of most cells and facilitate transmembrane glucose transport [2]. Thirteen GLUT subtypes are described with GLUT-1 and GLUT-3 most commonly expressed on cancer cells [2]. Upon entering the cell, [^18^F]FDG undergoes an initial phosphorylation reaction via hexokinase. Structural modifications produced by the hexose-[^18^F]FDG bond prevent its catabolism or extracellular transport at a high rate via glucose-6-phosphatase, hence metabolically “trapping” [^18^F]FDG [3].

The detection of [^18^F]FDG uptake relies on the ability of the PET detector to detect the natural radioactive decay of fluorine-18 attached to this glucose analogue. This occurs by beta+ decay which involves the conversion of fluorine to oxygen, releasing a positron which travels approximately 1mm before colliding with an electron, becoming neutralised and undergoing an annihilation reaction, producing a pair of gamma rays emitted at 180° from each other. These are detected by the scanner which draws a line of response between the rays. The crossing point of several lines of response indicates the area of greatest [^18^F]FDG uptake. Abnormal regions of [^18^F]FDG uptake are detected by comparison with the low overall background activity [3]. Standard uptake value (SUV) is a measure of [^18^F]FDG uptake by tissues and this is calculated by the division of the activity detected in the region of interest by the injected dose per unit body weight [3].

[^18^F]FDG competes with glucose for uptake by metabolically active tissue. It localises in tissues with a greater metabolic rate such as tumours, brain, salivary glands, myocardium, gastrointestinal tract, bladder, thyroid and gonads. [^18^F]FDG uptake has also been noted in brown adipose tissue in 2.3–4% of patients [3]. Thus, PET is not specific to cancerous tissues but those tissues with a greater than average metabolic rate offering functional information about the metabolic state of tissues. The advent of PET-CT has made it feasible to visualize the anatomical and metabolic properties of the tumour simultaneously.

### 1.3. Limitations: False Positives and Negatives

[^18^F]FDG-PET scanning is not without limitations, in particular false positives and false negatives impacting on sensitivity and specificity. False positives occur in tissues more metabolically active than background tissue such as inflammatory foci, limiting the specificity of [^18^F]FDG-PET. False positives can result in additional investigation, inappropriate treatment and altered clinical management with increased costs. Pancreatic cancer, commonly detected at a late stage, lacks an accurate method of detection often hindered by the difficulty in differentiating pancreatitis from pancreatic cancer and compounded by pancreatitis often accompanying cancer. Kubota et al. revealed that 24% of [^18^F]FDG uptake in pancreatic cancer is consumed by local inflammatory cells, demonstrating the poor specificity of [^18^F]FDG-PET in differentiating pancreatitis from cancer [4].

Thoracic diseases including tuberculomas, sarcoidosis, cryptococcosis and radiation fibrosis are a common source of false positive results, necessitating further invasive testing [3]. Immunosuppressed cancer patients or recipients of prior radiation are at increased risk of suffering from one of these conditions, effecting the efficacy of [^18^F]FDG-PET in cancer follow up or diagnosis of lung metastases.

False negatives are another major limitation to [^18^F]FDG-PET scanning where appropriate scanning fails to detect malignancy. Some of the underlying biology underpinning these false negative results will be outlined later in this review.

### 1.4. Clinical Importance

[^18^F]FDG-PET has a significant role in the diagnosis, staging, restaging, prognosis and monitoring of response to therapy in cancer. The functional information on tissue metabolism allows for identification of tumours otherwise undetectable on standard imaging modalities. PET has also had an impact on the treatment planning of cancer patients. Identification of occult metastases by PET alters the clinical management of patients, by avoiding futile surgery in favour of palliative interventions. Despite its usefulness, there is scope to improve the accuracy of PET by addressing the limitations outlined above.

## 2. Factors Affecting the Clinical Efficacy of PET

### 2.1. Gross Features

#### 2.1.1. Tumour Size

Small tumour size has been associated with decreased [^18^F]FDG uptake, accounting for false negative results in many studies [5,6,7,8,9]. This is most notable in the breast cancer setting where the persistent correlation between small tumour size and false negativity has meant that the National Comprehensive Cancer Network (NCCN) no longer endorses PET for evaluating stage I, II or operable stage III invasive disease [10]. A recent systematic review confirmed that [^18^F]FDG-PET does not have sufficient sensitivity to detect breast tumours <10 mm and is therefore not a recommended first line imaging modality for the initial assessment of primary breast tumours [5]. In bronchoalveolar lung cancer, a study with lung nodules <10 mm identified a negative PET scan in 20% of patients [8]. [^18^F]FDG uptake in cervical cancer is also influenced by size with a significant association between SUV and tumour size (r = 0.456, *p* = 0.025) [9]. In oral squamous cell carcinoma (SCC) higher T stage (3 and 4) have increased [^18^F]FDG uptake compared to lower T stages (1 and 2) [2]. Despite the evidence, small tumour size cannot exclusively cause false negatives. Higashi et al. identified a large 33 mm false negative tumour in their pancreatic study and highlighted that size alone does not influence decreased tumour [^18^F]FDG uptake [11].

#### 2.1.2. Tumour Grade

Aggressive lesions are associated with higher metabolism and increased [^18^F]FDG uptake compared to slow growing, less invasive types [5]. Tumour grade has been strongly associated with increased [^18^F]FDG uptake in breast, musculoskeletal and brain tumours however no correlation exists with mucinous tumours of the GI tract and lung [12,13]. In cervical cancer, Yen et al. reported higher [^18^F]FDG uptake in poorly differentiated, aggressive tumours compared to lower grade tumours in their 2004 study [9].

#### 2.1.3. Cellularity

The content of the tumour mass, in particular the cellular concentration has an association with [^18^F]FDG uptake. Poor [^18^F]FDG uptake by cystic mucinous, or signet ring tumours is attributed to fewer tumour cells forming the tumour mass [13]. Kim et al. and Higashi et al. have both reported a large difference in peak SUVs in mucinous bronchoalveolar carcinoma compared to cell-dense SCC or adenocarcinoma (AC) cancer types [14,15]. One large study by Higashi et al. revealed 57% of bronchoalveolar carcinoma patients as negative on PET while Berger et al. noted false negatives in 40% of patients with mucinous tumours [13,15]. Cellularity is also postulated to be linked to more rapid proliferation, increased likelihood of hypoxia and glycolysis, translating into increased [^18^F]FDG uptake [16].

### 2.2. Molecular Features

#### 2.2.1. Heterogeneity

The heterogeneity of the tumour microenvironment dictates the varied intratumoral uptake of [^18^F]FDG resulting in both false negatives and underestimations of tumour size [17]. Establishing these microenvironmental characteristics that influence [^18^F]FDG uptake is therefore important in order to optimise the clinical reliability of PET.

Currently, intratumoral variations in [^18^F]FDG uptake are not clearly defined in clinical practice when staging or planning radiation treatment in cancer, potentially misdiagnosing more extensive disease. Knowledge of tumour micro-environmental factors could augment PET efficacy with accurate prediction of tumour volume and disease extent with biomolecular profiles having a potential role [17].

#### 2.2.2. Metabolism

The preferential role for glycolysis over oxidative phosphorylation in cancer cells forms the basis of effective PET imaging. Glycolysis associated protein expression has been extensively studied with GLUT-1 and GLUT-3 overexpressed in a range of cancer types [18,19]. The evidence for correlating GLUT expression and [^18^F]FDG uptake is strong. Kurokawa et al. showed a positive correlation between [^18^F]FDG uptake and GLUT-1 expression [20]. Similarly Kunkel et al. associated high GLUT-1 to increased SUV in oral SCC [21] while Tian et al. confirmed this, no linear relationship between expression and [^18^F]FDG uptake was noted rather only overexpression facilitates increased SUVs [2].

The increased expression of GLUT-1 and GLUT-3 compared with other GLUT subtypes in cancer cells is in theory due to their significance in facilitating basal glucose transport in cells. They are vital in maintaining glycolysis despite a relative deficiency of glucose in the poorly perfused tumour microenvironment. GLUT-1 and GLUT-3 are therefore largely responsible for facilitating the Warburg effect, that is the preferential use of aerobic glycolysis by tumours compared with oxidative phosphorylation- and satisfying the abnormal glucose requirements of cancer cells. This provides some explanation for increased [^18^F]FDG uptake associated with these biomarkers of metabolism [2].

The metabolic profile of distant tumour metastasis differs from the primary tumour in some cases. Kurata et al. revealed elevated GLUT-3 and GLUT-5 in liver metastases compared to the primary lung cancer [22]. This differential expression of metabolic markers may present a limitation in biomolecular profiling for enhancing PET as some primary and secondary tumours appear to differ in their protein expression.

Despite several studies correlating high GLUT-1 and [^18^F]FDG uptake, some studies have revealed discrepant results postulating involvement of other metabolic factors. In oesophageal squamous cell cancer (SCC) hexokinase (HK) II had a higher correlation with [^18^F]FDG uptake than GLUT-1 expression [23]. Although not as prominent in influencing [^18^F]FDG uptake as GLUT, HK have been noted to strengthen the statistical significance of GLUT correlation with increased [^18^F]FDG uptake. Studies in oesophageal SCC and breast cancer could not demonstrate a significant correlation between tumour SUV and HK expression; on logistic regression however HK was identified as adding significance to the correlation between SUV and GLUT-1 expression [19]. This demonstrates the combined role of intracellular FDG transport via GLUTs with the commencement of glycolysis facilitated by HK.

The discordance in metabolic markers of [^18^F]FDG uptake between studies may not be an artefact but a source of biological significance. Correlations have been documented between GLUT expression and tumour aggressiveness as well as inflammation of normal tissue. Experimental models have shown GLUT-1 and 3 to be overexpressed in both tumour and inflammation though GLUT-1 was higher in tumour tissue while GLUT-3 was higher in inflammatory lesions [24]. This differential expression of biomarkers between cancer and inflammation, both of which exhibit increased [^18^F]FDG uptake on PET could potentially form some basis for differentiating benign from malignant lesions.

GLUT-1 appears to be the most prominently investigated GLUT in relation to [^18^F]FDG uptake. Though the studies described above have identified that GLUT-1 overexpression relates to increased [^18^F]FDG uptake, the correlation seems to vary between cancer types, something that may in part be attributable to the degree of tumour hypoxia (vide infra). Understanding and identifying cancers which exhibit the greatest correlation between GLUT-1 and [^18^F]FDG uptake could help highlight these cancers by means of a viable biomarker.

#### 2.2.3. Hypoxia

Hypoxia inducible factor 1-alpha (HIF-1α) is known to regulate glucose metabolism and consequently influence the regulation of [^18^F]FDG uptake. In the absence of oxygen, HIF-1α binds hypoxia response elements (HREs) causing expression of hypoxia responding genes related to angiogenesis, glycolysis and oxygen delivery. The underlying reasoning behind this mechanism is the cell’s attempt to prevent death; a mechanism manipulated by cancer in its expression of HIF-1α.

HIF-1α acts as an essential transcription factor involved in regulating metabolic functions by targeting a number of metabolism related proteins (Table 1) [25]. In addition, HIF-1α influences the relative contribution from metabolic function to overall energy production depending on the hypoxic state of the cell. Thus, the tumour hypoxic state affects the relative uptake of [^18^F]FDG. Pugachev et al. showed a positive correlation between [^18^F]FDG uptake and pimonidazole staining which identified hypoxic areas of tumour [17]. This supports the Dearling et al. study which revealed [^18^F]FDG uptake 1.26 times higher in hypoxic tumour regions versus normoxic areas [26].

Tumour necrosis has also been positively associated with [^18^F]FDG uptake in breast cancer [19]. This is probable due to the pre-necrotic hypoxic environment activating glycolysis and increasing [^18^F]FDG uptake. This theory is supported by pre-necrotic changes in cancer demonstrating increased [^18^F]FDG uptake.

#### 2.2.4. Angiogenesis

Tumour blood vessel status including microvascular blood volume (measured on functional MRI) and microvessel density has also been associated with variations in [^18^F]FDG uptake [19,27]. Though few studies exist regarding blood flow distribution in cancer, the evidence reveals that blood flow varies with the site, size, type of tumour and micro-vessel density [17]. Histologically identical tumours can also vary in their rates and distribution of blood flow [28]. The impact of angiogenesis on [^18^F]FDG uptake has been established in few cancer types such as breast cancer and malignant glioma where micro-vessel density and microvascular blood volume have been associated with increased [^18^F]FDG uptake [19,27].

The influence of angiogenesis on [^18^F]FDG uptake has led to investigation of the potential role of [^18^F]FDG PET in monitoring response to anti-angiogenic therapies such as bevacizumab. De Bruyne et al. demonstrated that low [^18^F]FDG uptake following bevacizumab therapy was associated with improved progression free survival in metastatic colorectal cancer [29]. Additionally, Colavolpe et al. showed that low [^18^F]FDG uptake on PET following bevacizumab treatment for glioma predicted longer progression-free survival, postulating reduced tumour angiogenesis, resulting in lower SUVs [30]. Similarly Goshen et al. concluded that pre and post bevacizumab [^18^F]FDG PET was superior in predicting pathological response to bevacizumab compared to standard restaging CT for metastatic colorectal cancer [31].

### 2.3. Interplay of Biological Features

Identifying individual biological features that influence [^18^F]FDG uptake is complicated by many of the biological processes being intricately linked. HIF-1α, expressed in hypoxia, regulates several key genes involved in angiogenesis and metabolism. Additionally, angiogenesis and metabolism influence each other and in turn have an impact of hypoxia.

For cellular [^18^F]FDG uptake to occur, it must reach the tumour site, making perfusion of the tumour facilitated by angiogenesis a key feature in controlling [^18^F]FDG metabolism [23]. Furthermore, metabolic demands in cancer influence hypoxia which induces vascular endothelial growth factor (VEGF) expression, facilitating blood vessel formation and tumour perfusion. Rapidly proliferating cells also require increased levels of glucose and differentiation has been correlated to [^18^F]FDG uptake [12,32,33]. When metabolic needs go unaddressed necrosis occurs causing inflammation with additional glucose demands and hypoxia. This intricate interplay between biological features highlights factors influencing [^18^F]FDG uptake are multifactorial and complex with interpretation of their combined effect on [^18^F]FDG uptake more important than any individual feature. Figure 1 illustrates this complex interplay of cellular processes.

### 2.4. Other Factors

#### 2.4.1. P-glycoprotein

Elevated expression of P-glycoprotein has been associated with decreased [^18^F]FDG uptake [34]. In hepatocellular carcinoma (HCC), both in vivo and in vitro models showed decreased [^18^F]FDG uptake with increased P-glycoprotein expression [35]. This suggests that [^18^F]FDG was a substrate of this drug efflux pump, with high levels of expression leading to reduced [^18^F]FDG uptake. Decreased [^18^F]FDG uptake and associate high P-glycoprotein expression have been observed in lung cancer and cholangiocarcinoma patients [34].

#### 2.4.2. Tumour Suppressor Genes

Tumour suppressor gene expression has been shown to influence SUVs. Vousden et al. demonstrated that p53 plays a significant role in cell metabolism and other essential cellular functions [36]. As p53 is mutated in up to 50% of tumours, and wild type p53 is anti-Warburg, promoting mitochondrial oxidative phosphorylation, the impact of p53 in promoting glycolysis is consequently potentially of great importance. Several studies have observed that variations in [^18^F]FDG uptake have been associated with mutated p53 [37,38]. In breast cancer, tumours with p53 mutations exhibit higher SUVs than those expressing the wild type protein [37]. In lung cancer, a statistically significant difference in [^18^F]FDG uptake was noted between cancers with no mutated tumour suppressors (Rb, P16, P27 and P53) and cancers with alterations which exhibited higher uptake values [38].

#### 2.4.3. Patient Factors

Multiple patient related factors are known to cause variable [^18^F]FDG uptake [39]. Patient size and body composition affects distribution of [^18^F]FDG and this is important considering the increasing prevalence of obesity and obesity-associated cancers [39]. Furthermore, high plasma glucose levels reduce [^18^F]FDG uptake [23]—an issue to account for in the diabetic and pre-diabetic setting. It is proposed that decreased [^18^F]FDG uptake is a result of the high glucose levels competing with [^18^F]FDG for cellular uptake [40]. The significance of glucose levels on FDG uptake remains controversial—while some studies report a significant effect of glucose levels on SUV, others dispute this [40,41,42,43]. The introduction of a ‘glycaemia modified SUV’ has been proposed, though there is no evidence of a linear relationship between glycaemia and SUV [11].

As blood glucose levels influence [^18^F]FDG uptake, drugs that can alter these levels need to be considered. In diabetics, drugs such as insulin or metformin, their dosage and administration time from commencement of PET scan could affect the reliability of PET. Consequently, cancer patients with comorbidities and drugs used to treat them can play a role in influencing [^18^F]FDG uptake. Corticosteroids may also affect [^18^F]FDG uptake. Zhao et al. compared the effect of prednisolone therapy on [^18^F]FDG uptake in granuloma and cancer xenograft rat models identifying that corticosteroids decreased [^18^F]FDG uptake in the granuloma models but not the cancerous lesions [44]. The potential that corticosteroids could help differentiate between inflammatory and cancerous lesions needs to be further validated as it could enhance the accuracy of PET scanning.

## 3. Optimising PET with Biomolecular Profiling

### 3.1. Stratification

It is evident from the literature that PET is a clinically useful diagnostic and prognostic imaging modality in oncology. However, variations in [^18^F]FDG uptake between patients highlight the apparent influence of tumour biology on PET accuracy and thus its efficacy. As stated herein, gross and molecular tumour features in addition to inherent patient characteristics play a role in influencing [^18^F]FDG uptake. If the relative influence of each of these factors could be ascertained both clinically and molecularly, it could be employed to enhance the accuracy of PET. The addition of biomolecular testing to PET imaging could also improve the sensitivity in identifying certain tumours. In an era where multimodal therapy is becoming increasingly utilized, the improved information obtained on the tumour could facilitate development of diagnostic algorithms for stratification of patients into appropriate treatment regimens.

This theory has been trailed by Hoeben et al. who investigated the significance of biomolecular profiling in mouse xenograft models, aiming to determine if combining immunohistochemistry (IHC) and [^18^F]FDG-PET parameters could reliably stratify Head and Neck cancers (HNC) into clusters [45]. By using [^18^F]FDG-PET as a biomarker and adding an IHC criterion, this group aimed to enhance prognostic prediction and facilitate appropriate treatment selection. Using 14 HNC lines grafted into mice, they revealed a distinct selection of biomarkers related to metabolism, proliferation, hypoxia and perfusion that could match tumours consistently to the correct cell line with high reliability [45]. The potential of combining [^18^F]FDG-PET with biomolecular profiling added value in terms of providing diagnostic and prognostic information.

### 3.2. Diagnosis and Predicting Prognosis

[^18^F]FDG avidity on PET is in itself a ‘biomarker,’ with studies citing it as a predictor of prognosis at diagnosis and post treatment in head and neck cancer (HNC) and oesophageal cancer [46,47,48]. In HNC pre-treatment high [^18^F]FDG uptake is associated with poor survival [49]. Conversely, HNC with an increased [^18^F]FDG pre-treatment had a better response to radiotherapy [46].

In oesophageal cancer, increased [^18^F]FDG uptake is also associated with poor prognosis compared to low [^18^F]FDG uptake [47,48]. Studies have also revealed however that SUVmax is nota prognostic parameter [50,51].

[^18^F]FDG uptake in oesophageal cancer is also identified as a predictor of lymph node disease, disease free survival (DFS) and recurrence [50,51]. A significant correlation between [^18^F]FDG uptake and tumour recurrence has also been noted in other cancer types [23,52,53,54]. The prognostic potential of [^18^F]FDG uptake in combination with biomarkers of tumour metabolism has also been evaluated. In oral SCC associations were found between increased [^18^F]FDG uptake in combination with increased GLUT-1 expression and poorer survival while another study showed similar results with GLUT-3 [21,55].

As described above, FDG uptake has been proposed as a prognostic biomarker in some small HNC and oesophageal cancer studies. Whether increased SUVs in smaller tumours predict a worse prognosis compared to decreased SUVs in larger tumours is not clearly defined from this research. It appears that several factors are responsible for predicting outcomes in combination with [^18^F]FDG uptake. By identifying molecular features that affect [^18^F]FDG uptake for individual cancers, biomolecular profiling could advance the role of PET in stratifying tumours and increase its efficacy. Table 2 outlines biomarkers and their associated influence on [^18^F]FDG uptake published to date.

## 4. Profiling Specific Cancer Types

The relationship between [^18^F]FDG uptake and tumour biology is not clearly defined with conflicting results between cancer types and subtypes with no definite consensus on which biomarkers are relevant for specific cancer types.

Discordance between studies regarding PET biomarkers can be attributed to variations in study design. Higashi et al. demonstrated in their pancreatic study that the numerical value of SUVs varied between different studies and between PET machines [11]. They suggest that SUV should not be used as an absolute value in the evaluation of [^18^F]FDG uptake rather broader categories of positive or negative uptake results are more important than absolute values [11].

Biological features and technical differences have both caused discordance in establishing the most appropriate and reliable biomarkers in relation to PET. However, there is evidence for specific biomarkers to predict levels of FDG uptake in some cancer types which are outlined below.

### 4.1. Oesophageal Cancer

PET’s ability to identify metastases not detected by conventional workup have been highlighted in oesophageal cancer [52]. Prediction of prognosis based on tumour SUVs has been shown with pathologic response and DFS correlated with SUV changes following induction therapy [53,54]. Importantly, 10–20% of oesophageal cancers are [^18^F]FDG negative on PET, demonstrating the need for biomolecular profiling to help identify this sub-group of tumours [56].

Potential biomarkers include size and GLUT-1 expression which positively correlate with SUV [23]. Taylor et al. could not identify a correlation between several prominent tumour markers and SUVmax, namely EGRF, P53, cyclin D1 and VEGF [57]. Although Schreurs et al. observed a significant relationship between HKII and SUVmax there were no significant relationships between GLUT-1, HK-1, HIF-Iα 1, VEGF-C, p53 and Ki-67 with SUV [58].

### 4.2. Breast Cancer

PET is not recommended for staging or follow-up in operable breast cancer as per the NCCN guidelines on account of the high rate of false negatives, largely attributed to small tumour size [10,59]. As a result, PET is not available to all breast cancer patients. Development of biomolecular profiles could help increase its accuracy by identifying tumours with likely poor [^18^F]FDG uptake. Recommended markers which indicate increased [^18^F]FDG uptake are GLUT-1 and HK-1 [19].

### 4.3. Non-Small Cell Lung Cancer (NSCLC)

The histological differences between NSCLC subtypes have revealed varied results in relation to PET accuracy. Several studies have suggested a high frequency of false negatives in bronchoalveolar carcinoma is due to decreased cellularity an assertion disproved by Yap et al. They showed sensitivity of [^18^F]FDG PET in bronchoalveolar carcinoma to be high overall with the introduction of more precise classification guidelines by the World Health Organisation [60]. GLUT-1, GLUT-3 and Ki-67 are potential biomarkers which have been positively correlated with increased [^18^F]FDG uptake in NSCLC [61].

### 4.4. Glioma

A SR on glioma revealed that [^18^F]FDG-PET has a sensitivity of 0.77 (95% CI, 0.66–0.85) and specificity of 0.78 (95% CI, 0.54–0.91) [62]. The only biomarker identified influencing [^18^F]FDG uptake is VEGF [63]. The practicality of using biomolecular profiling to increase sensitivity is made difficult by the inability to obtaining tumour samples as these tumours are often unresectable.

### 4.5. Head and Neck Cancer

Gronroos et al. revealed that an increased [^18^F]FDG uptake is associated with a more aggressive phenotype and therefore high P53 and VEGF expression [64]. Detecting the presence or absence of these markers in HNC could enhance accuracy of PET. A recent study by Rasmussen et al. reported a positive correlation between SUVmax and β-tubulin-1 index and significant negative correlations between SUV max and Bcl-2 and P16 [65].

## 5. Conclusions and Future Directions

The influence of biomolecular markers on [^18^F]FDG uptake has been established and is clearly linked with hypoxia, metabolism and angiogenesis. Despite this, a definite consensus is lacking on associations between biomarkers, [^18^F]FDG uptake and cancer. Considerable variation and heterogeneity in study design including small sample size, variation in PET algorithms employed between centres; the diverse molecular markers examined and the lack of validation are clearly an issue limiting firm conclusions, and further research is clearly warranted. Biomolecular profiling can articulate the true significance of [^18^F]FDG uptake while also addressing the limitations of PET in clinical oncology such as false negative results. There is a compelling case that the integration of biomolecular profiling and [^18^F]FDG PET could enhance diagnosis, improve prognosis prediction and facilitate appropriate stratification of patients to treatment regimens based on a clear characterisation of the tumour, but this needs to be validated in rigorous scientific study.

## Figures and Tables

**Figure 1 pharmaceuticals-12-00016-f001:**
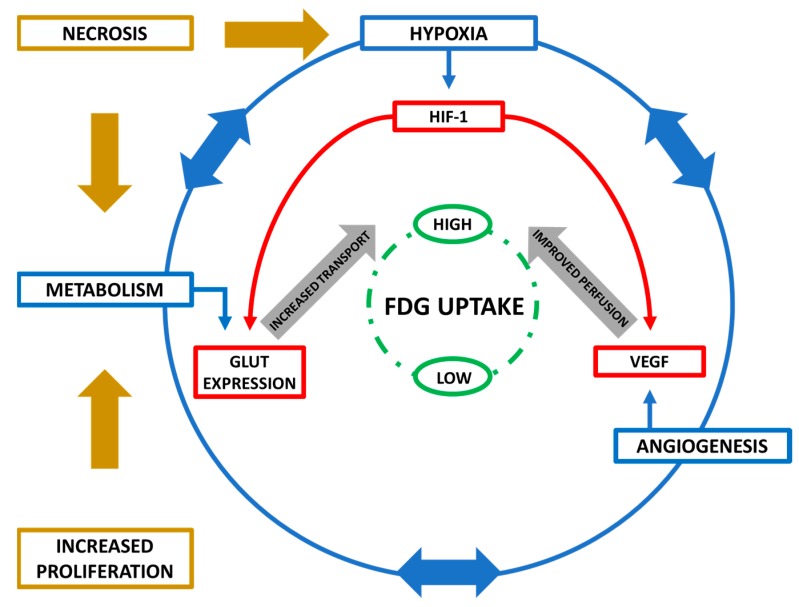
Biomolecular influences on [^18^F]FDG uptake. Metabolism, hypoxia and angiogenesis all play a role in glucose and therefore [^18^F]FDG uptake via their associated biomolecular proteins (GLUT, HIF-1α and VEGF respectively). Interrelationships exist between metabolism, hypoxia and angiogenesis such that they play a role in regulating each other. Proliferation and necrosis-induced inflammation increase overall tumoral energy requirements, also driving metabolism and contributing to this complex network.

**Table 1 pharmaceuticals-12-00016-t001:** Targets of HIF-1α. Adapted from Denko et al. [25].

Target Genes	Metabolic Function
GLUT-1/GLUT-3	Cellular Glucose Entry
HKII	Phosphorylation
PGI, PFK1, Aldolase, TPI, GAPDH, PGK, PGM, enolase, PK, PFKFB1-4	Glycolysis
LDHA	Pyruvate>Lactate Conversion
MCT4	Cellular Lactate Removal
PDK1, MXI1	Decreased Mitochondrial Activity
COX4I2, Lon Protease	O_2_ Consumption in Hypoxia

GLUT: Glucose uptake transporter; HK: Hexokinase; PGI: Glucose-6-phosphate isomerase; PFK: Phosphofructokinase; TPI: Triose phosphate isomerase; GAPDH: Glyceraldehyde 3-phosphate dehydrogenase; PGK: phosphoglycerate kinase; PGM: phosphoglucomutase; PK: pyruvate kinase; PFKFB: 6-phosphofructo-2-kinase/fructose-2,6-biphosphatase; LDHA: Lactate Dehydrogenase A; MCT: Monocarboxylate transporter; PDK: Pyruvate dehydrogenase kinase; MXI: MAX-interacting protein; COX: Cytochrome c oxidase.

**Table 2 pharmaceuticals-12-00016-t002:** Metabolic, hypoxic and angiogenic biomarkers affecting [^18^F]FDG uptake in different cancer types. A positive association (+) indicates [^18^F]FDG uptake increased with biomarker. A negative association (−) indicates [^18^F]FDG uptake decreased with biomarker. A null association (0) indicates biomarker expression was unrelated to [^18^F]FDG uptake.

Cancer Type	[^18^F]FDG Uptake Association	Biomarker	Function	Reference
Oesophageal SCC	+ + − + 0 0	HK-I HK-II * HK-II VEGF VEGF KI67	Metabolism Metabolism Metabolism Angiogenesis Angiogenesis Proliferation	[19] [19] [58] [66] [57,66] [23]
Oesophageal AC	+ − 0 0 0 0	GLUT-1 HK-II HIF-1α VEGF P53 Ki67	Metabolism Metabolism Hypoxia Angiogenesis TSG Proliferation	[23,57] [58] [58] [23] [58] [58]
Breast	+ + 0 0 0 0	GLUT-1 HK-1 HK-II ** HK-III HIF-1α VEGF	Metabolism Metabolism Metabolism Metabolism Hypoxia Angiogenesis	[19] [19] [19] [19] [19] [19]
Head and Neck	− + +	GLUT-1 GLUT-3 VEGF	Metabolism Metabolism Angiogenesis	[2,21,64] [2] [64]
Oral SCC	+ + + +	GLUT-1 ** GLUT-3 ** HK-II HIF-1α	Metabolism Metabolism Metabolism Hypoxia	[2,21,67] [2,21] [67] [67]
Cervical	+ +	GLUT-1 HK-II	Metabolism Metabolism	[9,68] [68]
Pancreatic	+	GLUT-1	Metabolism	[18]
Ovarian	+	GLUT-1	Metabolism	[20]
NSCLC	+ + 0 + 0	GLUT-1 GLUT-3 GLUT-3 HIF-1α Ki-67	Metabolism Metabolism Metabolism Hypoxia Proliferation	[61,69] [61] [69] [69] [69]
Glioma	+	VEGF	Angiogenesis	[63]
Gastric	0 0 + 0	GLUT-1 HKII HIF-1α PCNA	Metabolism Metabolism Hypoxia Proliferation	[70] [70] [70] [70]
Colorectal	+ 0	HIF-1α PCNA	Hypoxia Proliferation	[71] [71]
Musculoskeletal	+ +	GLUT-1 HK-II	Metabolism Metabolism	[72] [72]
Hodgkin’s Lymphoma	+ 0 0	GLUT-1 GLUT-3 HK-II	Metabolism Metabolism Metabolism	[73] [73] [73]
Thyroid	0 0 0 +	GLUT-1 GLUT-3 HK-II VEGF	Metabolism Metabolism Metabolism Angiogenesis	[74] [74] [74] [74]

* No significant correlation between these biomarkers and SUV though in logistic regression they added value to GLUT-1 correlation. ** These biomarkers were only found to correlate with increased SUV when overexpressed.

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
