# Peer review of "Can the Efficacy of [^18^F]FDG-PET/CT in Clinical Oncology Be Enhanced by Screening Biomolecular Profiles?"

_pharmaceuticals, 2019, doi:10.3390/ph12010016_

Round 1
Reviewer 1 Report
Interesting and well structured review but the reference list could be improved.
Maybe the title should include FDG, this would make clear that the emphasis is on FDG PET
and not PET in general.
Please stick to the nomenclature rules for radiopharmaceuticals which can be found e.g. here Nucl Med Biol. 2017 Dec;55:v-xi. doi: 10.1016/j.nucmedbio.2017.09.004
Line 60: it should be beta+ decay instead of beta decay to describe the decay process.
Line 107: ...many studies. Please indicate. I miss the references to these studies.
Line 109: ...(NCCN) no llonger endorses PET... again, please add the reference.
Line 124: please add an example for those poorly differentiated aggressive tumours.
Line 133: please add reference.
Line 135-144: please add references.
Line 181: please add references.
Line 213: please add references.
Figure 1: Format of capture. Please check.
Line 251: please provide examples for cancer types associated with PGP expression decreased uptake. Could improve understanding.
Line 257: please add reference.
Line 263: please add reference.
Line 267: please add reference.
Line 268: please add reference.
Line 299: please add reference.
Line 303: shouldn´t it be ...FDG is in itself a biomarker....? FDG-PET is the technique to determine FDG not the biomarker...
Line 304: please add reference.
Line 317: please add reference.
Line 317-324: please add references.
Line 331: please add reference.
Line 336: please provide examples for those cancer types.
Line 337: please add reference.
Line 345: blank too much
Line 350. please add reference.
Line 369: please add reference.
Line 390: FDG-PET
Author Response
See attached word document

Reviewer 2 Report
1)Page 2, line 44- the non-specific tracers listed should be checked. One must specify the radiotracer form as for eg: natural choline is a PET tracer, neither is fluoride. Also, it is C-11 acetate and not C-11-acetate.
2) Page 2, line 80- I assume you mean false positives here when you say “The false detection of malignant tissue….” Would be helpful to re-word this sentence to convey the point.
3)Page 3, line 86 and line 118- Both use ref 4 but from the list of references 4 is not Kubota et al therefore use of ref 4 in line 86 might be incorrect. Please check the reference.
4) Page 3, line 111- Is FDG not useful for detection of primary tumors less than 10mm or all primary tumors?
5) Page 3 and 4- section on cellularity- Please check if reference 9 used here is correct.
6) Page 4, line 168- Please clarify how HK has strengthened the association GLUT with FDG. In particular, please specify the relation of HK to GLUT.
7) Page 6, line 216- High micro-vessel density or high blood flow to the tumor increases FDG? Can FDG be used as a response to therapy for anti-angiogenic drugs like avastin? What about effects of EPR (enhanced permeability and retention) on FDG uptake?
Very well written manuscript. Interesting read and useful.
Author Response
See attached word document.

Round 2
Reviewer 1 Report
-